# Collagen XII Deficiency Increases the Risk of Anterior Cruciate Ligament Injury in Mice

**DOI:** 10.3390/jcm10184051

**Published:** 2021-09-07

**Authors:** Shin Fukusato, Masashi Nagao, Kei Fujihara, Taiju Yoneda, Kiyotaka Arai, Manuel Koch, Kazuo Kaneko, Muneaki Ishijima, Yayoi Izu

**Affiliations:** 1Department of Medicine for Orthopaedics and Motor Organs, Juntendo University Graduate School of Medicine, 2-1-1 Hongo, Bunkyo-Ku, Tokyo 113-8421, Japan; s-fukusato@juntendo.ac.jp (S.F.); k-kaneko@juntendo.ac.jp (K.K.); ishijima@juntendo.ac.jp (M.I.); 2Medical Technology Innovation Center, Juntendo University, 2-1-1 Hongo, Bunkyo-Ku, Tokyo 113-8421, Japan; 3Graduate School of Health and Sports Science, Juntendo University, 1-1 Hiragagakuenndai, Inzai 270-1695, Japan; 4Department of Laboratory Animal Science, Faculty of Veterinary Science, Okayama University of Science, 1-3 Ikoinooka, Imabari 794-8555, Japan; v18m122fk@ous.jp (K.F.); v18m147yt@ous.jp (T.Y.); 5Department of Veterinary Surgery, Faculty of Veterinary Science, Okayama University of Science, 1-1 Ridai-cho, Kita-ku, Okayama 700-0005, Japan; k-arai@vet.ous.ac.jp; 6Institute for Dental Research and Oral Musculoskeletal Biology and Center for Biochemistry, Medical Faculty, University of Cologne, D-50931 Cologne, Germany; manuel.koch@uni-koeln.de

**Keywords:** collagen XII, ACL injury, risk factor

## Abstract

Anterior cruciate ligament (ACL) rupture is a common knee injury for athletes. Although surgical reconstruction is recommended for the treatment of ACL ruptures, 100% functional recovery is unlikely. Therefore, the discovery of risk factors for ACL ruptures may prevent injury. Several studies have reported an association between polymorphisms of the collagen XII gene *COL12A1* and ACL rupture. Collagen XII is highly expressed in tendons and ligaments and regulates tissue structure and mechanical property. Therefore, we hypothesized that collagen XII deficiency may cause ACL injury. To elucidate the influence of collagen XII deficiency on ACL, we analyzed a mouse model deficient for *Col12a1*. Four- to 19-week-old male *Col12a1*^-/-^ and wild-type control mice were used for gait analysis; histological and immunofluorescent analysis of collagen XII, and real-time RT-PCR evaluation of *Col12a1* mRNA expression. The *Col12a1*^-/-^ mice showed an abnormal gait with an approximately 2.7-fold increase in step angle, suggesting altered step alignment. *Col12a1*^-/-^ mice displayed 20–60% ACL discontinuities, but 0% discontinuity in the posterior cruciate ligament. No discontinuities in knee ligaments were found in wild-type mice. Collagen XII mRNA expression in the ACL tended to decrease with aging. Our study demonstrates for the first time that collagen XII deficiency increases the risk of ACL injury.

## 1. Introduction

Injury to the anterior cruciate ligament (ACL) is a common trauma for athletes of pivot sports, such as soccer and basketball [1,2,3]. Surgical reconstruction is generally recommended because of the lack of knee joint stability, and this injury increases the risk of future osteoarthritis [4,5,6,7]. However, clinical treatment does not allow all athletes to return to their original sports level [1,6]. Thus, discovery of associated risk factors leading to ACL injury is required for prevention. 

Although the risks of ACL injury are significantly higher in athletes than nonathletes, a familial association has been indicated [8,9,10]. Various genetic variants are associated with ACL injury. One of these gene variants comprises single nucleotide polymorphisms (SNPs) in the collagen type XII alpha 1 chain (*COL12A1*) gene, and *COL12A1* SNPs have been found worldwide, including in China [10], India [11], Poland [12], and South Africa [3]. Therefore, collagen XII may be a potential risk factor for developing an ACL injury. However, how *COL12A1* SNPs influence ACL function has not been elucidated.

Collagen XII belongs to the subfamily of fibril-associated collagens with interrupted triple helices (FACIT), and is an important regulator of fibrillogenesis and fiber organization in tendons and ligaments [13,14]. Collagen XII is produced in response to load and therefore highly expressed in load-bearing tissues. Genetic mutations in *COL12A1* cause myopathic Ehlers–Danlos syndrome (mEDS) [15,16,17,18], which exhibits hypermobility of distal joints and proximal joint contractures combined with congenital muscle hypotonia. A similar phenotype was also found in a mouse model with deletion of the *Col12a1* gene [14,17,19]. These lines of evidence indicate a possible function of collagen XII in ACL. Therefore, we hypothesized that the collagen XII deficiency may cause ACL injury. The aim of the present study was to elucidate the influence of collagen XII on ACLs and explore the potential of collagen XII as a risk factor for ACL injury using the *Col12a1* deficient mouse model. 

## 2. Materials and Methods

### 2.1. Animals

The generation of the *Col12a1* deficient mouse model used in this study has been previously described [14]. Male mice were used at an age of 4–19 weeks. All animal studies were approved and performed in compliance with the Institutional Animal Care and Use Committee (IACUC) at Okayama University of Science.

### 2.2. Gait Analysis

Video recordings of 15-week-old *Col12a1* deficient (*n* = 3) and wild-type control (*n* = 3) mice were conducted in an open area. In addition, the palms of the feet of each mouse were painted with water-based paint. The animals were placed in a transparent tunnel lined with a sheet of paper and allowed to walk extensively. Footprints were analyzed as previously described [20]. Briefly, the step angle between the left and right foot palms was measured (Figure 1a), and nine steps were collected from each mouse and the mean values were calculated.

### 2.3. Histological Analysis 

Five-week-old and 17–19-week-old *Col12a1* deficient (*n* = 5 and *n* = 15 respectively) and wild-type control mice (*n* = 4 and *n* = 8 respectively) were utilized for histological analyses. Mice were euthanized with an overdose of isoflurane. Hindlimbs were dissected out and fixed in 4% formaldehyde for 24 hours at 4 °C, followed by decalcification with 0.5 M ethylenediaminetetraacetic acid (EDTA; pH 8.0) for 14 days at 4 °C on a shaker. The tissues were embedded in paraffin and 2-µm sections were prepared along the sagittal plane. A standard protocol was used for hematoxylin/eosin (HE) and safranin O/fast green staining. Discontinuity of the ACL and posterior cruciate ligament (PCL) was determined using safranin O/fast green stained sections, and quantitative analysis was performed on either side of the hindlimb by blind selection. Images were captured using an Olympus BX53 microscope.

### 2.4. Immunofluorescence Analysis

The knee joints were dissected from 17-week-old wild-type mice (*n* = 3). Tissues were fixed with 4% paraformaldehyde, decalcified with 15% EDTA/0.01 M phosphate-buffered saline, embedded in OCT freezing medium, and stored at –80 °C. Sagittal sections (5 µm) were cut and used for immunofluorescence analyses. Immunofluorescence localization was performed as previously described [14] using a rabbit anti-type XII collagen antibody (1:1000 dilution; KR33, Manuel Koch, Cologne, Germany) and DAPI staining (for visualizing nuclei). Images were captured using a Carl Zeiss Axioplan 2 fluorescence microscope.

### 2.5. Real-Time RT-PCR Analysis

Total RNA was isolated from the ACLs and Achilles tendons of 4- (*n* = 4) and 17-week-old (*n* = 4) wild-type mice using the RNeasy Micro kit (Qiagen GmbH; Hilden, Germany) according to the manufacturer’s protocol. Reverse transcription was performed using 500 ng of total RNA with random primers (Applied Biosystems, Foster City, CA, USA). Quantitative real-time PCR was performed using QuantStudio3 (Applied Biosystems). The primer sequences were as follows: *Col12a1*, forward 5ʹ GTC ATA CAC TCT CAA ATT CCT CAC 3ʹ and reverse 5ʹ GAC ACT CCA TAC ACC ATC ACG 3ʹ; *Gapdh*, forward 5ʹ GTG GAG TCA TAC TGG AAC ATG TAG 3ʹ and reverse 5ʹ AAT GGT GAA GGT CGG TGT G 3ʹ. The PCR conditions were initially 95 °C for 20 s followed by 45 cycles at 95 °C for 3 s and 60 °C for 30 s. The mRNA expression levels were normalized to the relative levels of the housekeeping gene *Gapdh* using the comparative threshold (Ct) method.

### 2.6. Statistical Analysis

Graph Pad Prism 9 (GraphPad Software, Inc., La Jolla, CA, USA) was used for statistical analysis. Results are presented as means ± standard deviation. Statistical significance was determined using unpaired two-sided *t*-tests. Student’s *t*-test for step angle analysis and Fisher’s exact tests for incidence of ACL discontinuity were performed for comparisons. 

## 3. Results

### 3.1. Col12a1 Deficient Mice Demonstrate Knee Deformity with an Abnormal Gait

The *Col12a1*^-/-^ mice were generated previously by targeted deletion of exons 2–5 and muscle [17], bone [14], and the flexor digitorum longus tendons [19] were analyzed. To evaluate locomotion, the *Col12a1*^-/-^ and wild-type control mice were examined using the open field behavioral test under veterinary consultation (Figure 1b, Appendix A, Appendix A). Compared with the control mice, the *Col12a1*^-/-^ mice displayed a gait abnormality and appeared to have reduced mobility. A mild limitation of knee joint extension was palpable in the *Col12a1*^-/-^ mice.

We performed gait analysis by measuring the step angle between the left and right palms of the feet. The step angle measured in this study is shown in Figure 1b. The footprints of control mice were almost parallel to the direction of travel, whereas those of the *Col12a1*^-/-^ mice were tilted toward the outside (Figure 1b). Statistical analysis showed that the step angle was approximately 2.7 times wider in the *Col12a1*^-/-^ mice than in wild-type controls (Figure 1b–d). These results demonstrated that collagen XII deficiency altered step alignment.

### 3.2. Collagen XII Deficiency Causes ACL Discontinuity

To investigate the morphological changes in ACLs of *Col12a1*^-/-^ mice, histological analyses of knee joints from young (5-week-old) and mature (17–19-week-old) mice were performed using HE and safranin O/fast green staining. Microscopic imaging of knee joints demonstrated that wild-type ACLs and PCLs connected the tibia to the femur in both young and mature mice (Figure 2a). In contrast, although the ACLs of *Col12a1* deficient mice were detected in the space between the femur and tibia, some of the ACLs were discontinuous and displayed gaps within the ligament structure (Figure 2b). Interestingly, no discontinuity was observed in the PCLs from *Col12a1* deficient mice. The articular cartilage and meniscus were comparable between the genotypes and different age groups based on the safranin O/fast green staining. We performed quantitative analysis of ACL discontinuity in 5-week-old mice, which revealed that 0% of wild-type mice and 20% of *Col12a1* deficient mice demonstrated an ACL discontinuity (Table 1). Whereas in mature mice, 0% of wild-type mice and 53% of *Col12a1* deficient mice developed ACL discontinuity. The univariate analysis with Fisher’s exact tests in 19weeks old, but not 5 weeks, revealed the significant difference between *Col12a1* deficient and wild-type mice in mature stage. In contrast, quantitative analysis revealed 0% PCL discontinuity for both genotypes and age groups. These data indicate that collagen XII deficiency increases the risk of ACL injury.

### 3.3. Collagen XII Is a Component for the ACL

To explore whether collagen XII is expressed in the ACL, immunofluorescence analysis of knee joints from 17-week-old wild-type mice was performed using an antibody against collagen XII (Figure 3). Nuclei were visualized by DAPI staining. Collagen XII was detected in the ACL and displayed a fibril-like arrangement. These data indicate that collagen XII is one of the critical components for the ACL.

### 3.4. Collagen XII Expression Tends to Decrease in Knee Ligaments and the Achilles Tendon with Aging

The mRNA expression of collagen XII was measured in the ACLs and Achilles tendons obtained from 4- and 17-week-old mice by real-time RT-PCR (Figure 4). *Col12a1* mRNA was detected in both tissues and the expression levels were comparable between the ACL and Achilles tendon in 4-week-old mice. *Col12a1* mRNA expression tended to decrease in both tissues in the 17-week-old mice compared with the younger mice; however, no significant differences were detected. These data indicate that collagen XII quantitatively regulates tendon and ligament functions.

## 4. Discussion

In the present study, we demonstrated that collagen XII deficient mice exhibited gait abnormalities with increased step angle and increased development of ACL discontinuity. Collagen XII was expressed in ACL, and its mRNA expression level tended to decrease with age. These lines of evidence indicated that collagen XII is the essential component of ACL and a loss of collagen XII is a risk factor for ACL injury. 

Collagen XII is α1 homotrimer, which consists of a large NC3 domain and a short collagen helix interrupted by a NC2 and NC1 domain [13]. The *COL12A1 AluI* RFLP polymorphism associated with ACL injury is located within exon 65, which encodes the NC1 domain [3,10,11,12,21]. The C-terminal including collagenous and NC1 domains are important for binding to decorin, fibromodulin [22], and cartilage oligomeric matrix protein (COMP) and thereby holding these on the surface of collagen fibrils. Interestingly, the sequence variants in *DCN* and *FMOD* are also associated with ACL injury [23]. In addition, the extracellular decrease of decorin was found in the cell culture of dermal fibroblasts obtained from mEDS patients [24], and the mice deficient for decorin or fibromodulin demonstrate abnormal fibrillar organization and decreased mechanical properties in tendons and cornea [25,26,27,28]. These data suggest that collagen XII regulates fibrillogenesis via interacting with other extracellular molecules, thereby maintaining tissue stability and mechanical strength. On the other hand, COMP is localized along with collagen XII in anchoring plaques in human skin and functions as adapter protein [22]. After ACL injury, the concentration of COMP in synovial fluid is increased, suggesting a disruption of the binding between collagen XII and COMP. These lines of evidence suggest that loss of collagen XII impairs molecular functions of its binding partners, resulting in altered ACL structure and mechanical properties, leading to ACL injury. 

The genetic variations in other collagens such as *COL1A1*, *COL3A1*, *COL5A1*, and *COL6A1* are also associated with ACL injury [9,10,21]. Particularly, *COL5A1* rs12772 and *COL12A1* rs9705647 variants were significantly associated with ACL injury. Both collagens V and XII are localized at the cell surface and have a critical role of fibrillogenesis and fiber organization [19,29,30], suggesting that impaired pericellular interaction may be associated with the cause of ACL injury. However, neither a direct nor an indirect molecular interaction between collagens V and XII has been identified. Further studies are required. On the other hand, collagens V and VI are functionally interacted in the cell-matrix environment [24,31]. Collagen VI- and XII-related disorders share clinical overlaps [17,18,24], and our previous study demonstrated a functional relationship between collagens VI and XII during osteogenesis [32]. Noteworthy, the mutations in *COL5A1*, *COL6A1*, or *COL12A1* cause connective tissue disorders in humans, altering fiber organization and mechanical properties, resulting in tissue fragility [15]. On the basis of these findings suggest that genetic variation in collagen genes may disrupt interactions necessary for fibrillogenesis and fiber organization that results in decreased mechanical properties leading to the ACL injury. However, further studies are required.

In addition to the reduced mechanical properties in *Col12a1* mutations, several studies demonstrated that tensile strain acting on cells regulates collagen XII production [33,34,35]. Together with our data on gait abnormalities and ACL discontinuity in *Col12a1*^-/-^ mice suggest that *Col12a1* deficient ACLs may have reduced mechanical strength against knee force. Our data also demonstrated that collagen XII expression levels tended to decrease with age, and the incidence of ACL injury in older *Col12a1*^-/-^ mice was greater than in the younger mice. The phenotype of the mEDS patients with reduces collagen XII amount is milder than complete loss, but gait abnormalities remain [24]. These lines of evidence indicate that collagen XII quantitatively regulates fibrillogenesis and tissue organization of the ACLs. Hence, the age-dependent quantitative change in collagen XII may be associated with the risk of ACL injury. 

Tendons and ligaments are structurally and functionally similar; however, there are slight differences in components, strength, and mechanics. Our study indicated that *Col12a1* mRNA expression level was comparable between ACLs and Achilles tendons, but we found no Achilles tendon ruptures in the *Col12a1*^-/-^ mice. This may be because that the ACL is much less susceptible for tensile strength than tendons [36], or due to differences in the mechanical forces applied to these tissues. In support of this, ACL injury-association polymorphisms of *COL12A1 AluI* and *BsrI* RFLPs were not detected in patients with Achilles tendon injury [3]. These lines of evidence suggest that collagen XII may play a precise regulatory role between tendons and ligaments. Further studies are required in the future. 

In conclusion, collagen XII was the critical component for ACLs and its absence was susceptible to ACL injury. Therefore, collagen XII levels may be useful as a risk factor for ACL ruptures in athletes, as well as for tendon/ligament injuries in older individuals. The current study will contribute to the establishment of a novel preventive strategy for joint disorders in the future.

## Figures and Tables

**Figure 1 jcm-10-04051-f001:**
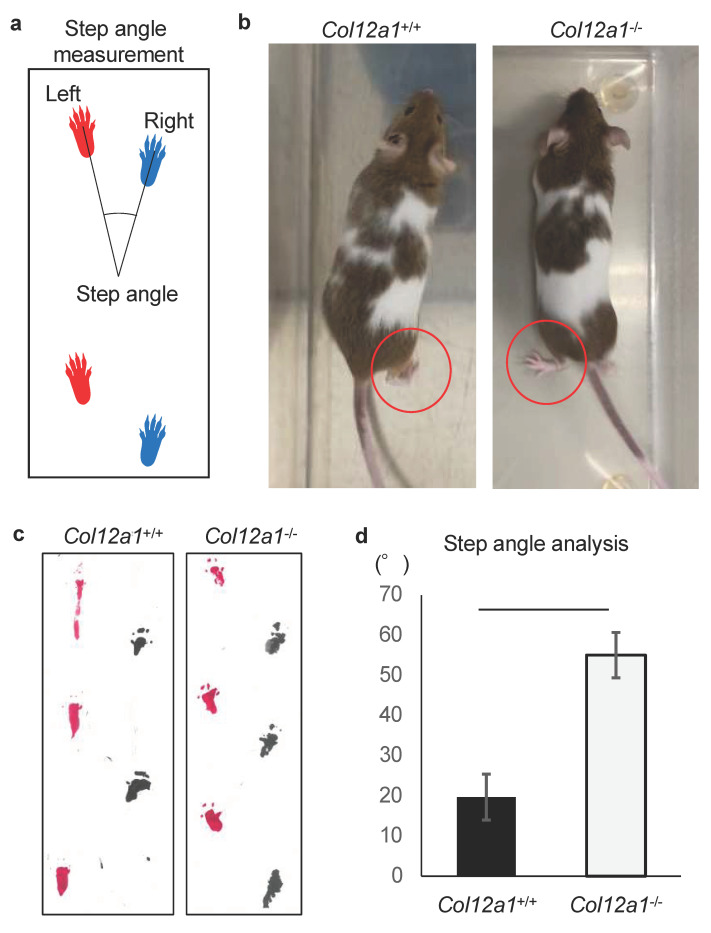
Gait abnormality in *Col12a1* deficient mice. (**a**) The method diagram for measuring the step angles from the footprints. (**b**) Images of gait analyses for wild-type control (*Col12a*^+/+^) and *Col12a1*^-/-^ mice at 4-week-old. The control mouse lifted its heel (red circle), whereas the heel of the *Col12a1*^-/-^ mouse was down, and the foot was positioned outward (red circle). (**c**) Representative footprints from wild-type control and *Col12a1*^-/-^ mice are shown. (**d**) The step angle was statistically larger in the *Col12a1*^-/-^ mice compared with wild-type controls (*p* < 0.01).

**Figure 2 jcm-10-04051-f002:**
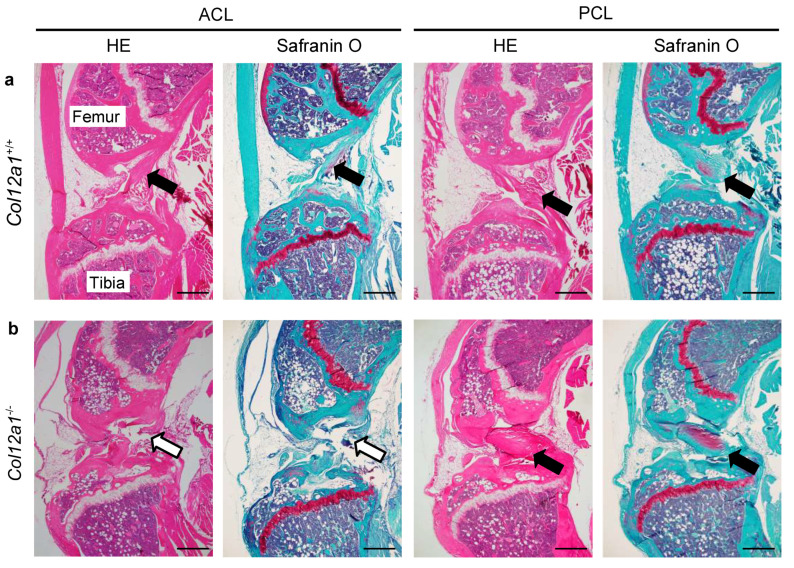
ACL discontinuity in *Col12a1* deficient mice. Hematoxylin/eosin and safranin O/fast green staining of sagittal sections of the knee joints from 19-week-old wild-type and *Col12a1*^-/-^ mice. In wild-type mice, ACLs and PCLs were found in the knee joints where they ran diagonally and connected the femur and tibia (**a**, black arrows). In contrast, in *Col12a1^-/-^* mouse knee joints, ACLs were discontinuous (**b**, open arrows). The *Col12a1*^-/-^ PCLs were intact, and no differences were observed between genotypes (**b**, black arrows). Bars = 500 µm.

**Figure 3 jcm-10-04051-f003:**
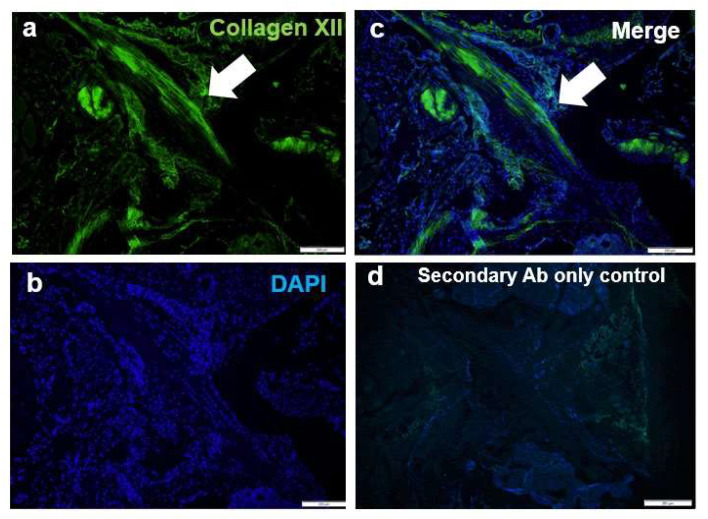
Collagen XII immunofluorescence in the ACL. Immunofluorescence of collagen XII was detected (arrows) in ACLs of 17-week-old wild-type mice. (**a**) Collagen XII is represented by green fluorescence. (**b**) Nuclei were visualized by DAPI staining (blue). (**c**) Merged images. (**d**) Secondary antibody only was used as a staining control. Bars = 200 µm.

**Figure 4 jcm-10-04051-f004:**
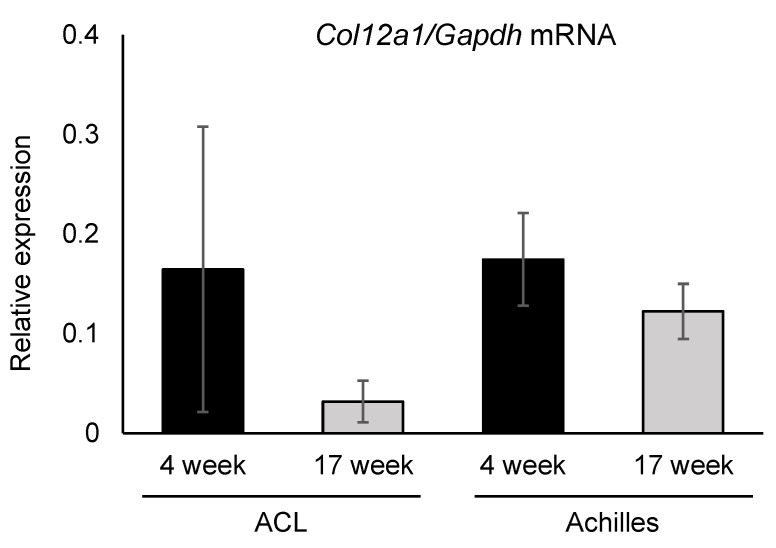
Expression of *Col12a1* mRNA in the ACLs and Achilles tendons. The expression of *Col12a1* mRNA normalized to *Gapdh* was measured. The *Col12a1* mRNA expression level was comparable between the ACL and Achilles tendon in 4-week-old mice. The expression levels tended to decrease at 17 weeks compared with 4 weeks of age, but the differences were not statistically significant.

**Table 1 jcm-10-04051-t001:** Incidence of ACL discontinuity in young and mature *Col12a1*^-/-^ and *Col12a1*^+/+^ control mice.

	Young Stage (5-Week-Old)	Mature Stage (17–19-Week-Old)
*Col12a1* ^+/+^	0% (0/4)	0% (0/8)
*Col12a1* ^-/-^	20% (1/5)	53% (8/15)

## Data Availability

Data analyzed within this study are included in this body of the manuscript. Data are also available from the corresponding author upon request.

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
