# Peer review of "Collagen XII Deficiency Increases the Risk of Anterior Cruciate Ligament Injury in Mice"

_jcm, 2021, doi:10.3390/jcm10184051_

Round 1

Reviewer 1 Report

My comments to the authors are:

In general, the manuscript must be rewritten, including all sections, not repeating information, and each section with its text. In addition, the authors should make an effort to better explain the rationale for the topic and relate their results to previous studies.

The manuscript needs to be checked by a native English-speaking person.

The abstract should have the sections that are in the journal guide. The objective and the methodology are not clear

The introduction should be clearer. Authors must further explain the studies related to the topic to understand the justification of their paper.

How many animals did you use for the study?

The description of the statistical analysis should be more complete

In the results section, the authors should explain only their data. They should not explain how they did it (that is the method) or give their opinion (that is under discussion). Authors must rewrite this section

Authors must include limitations and conclusion sections

The reference style is correct

Author Response

In general, the manuscript must be rewritten, including all sections, not repeating information, and each section with its text. In addition, the authors should make an effort to better explain the rationale for the topic and relate their results to previous studies.

  • Thank you very much for your comments and suggestions. We sincerely accept your suggestions and rewrote whole manuscript.

The manuscript needs to be checked by a native English-speaking person.

  •  We use Edaz English editorial services (https://jp.edanz.com/ac) and whole manuscript is edited by Susan Zunino, PhD, from Edaz. We include this in the acknowledgement.

The abstract should have the sections that are in the journal guide. The objective and the methodology are not clear

  • We followed the following journal guideline; The abstract should be a total of about 200 words maximum. The abstract should be a single paragraph and should follow the style of structured abstracts, but without headings.
  • As the reviewer suggested, we added the following sentence to clarify the objective of this study; “Therefore, we hypothesized that collagen XII deficiency may cause ACL injury.”  

The introduction should be clearer. Authors must further explain the studies related to the topic to understand the justification of their paper.

  • We have revised as the reviewer suggested.

How many animals did you use for the study?

  • We rewrote the method to include a detailed number of animals for each experiment. Briefly, regarding to the table1, four wildtype and five Col12a1-/-mice were used for young stage analysis, and eight wildtype and 15 Col12a1-/- mice were used for the mature analysis.

The description of the statistical analysis should be more complete

  •  We rewrote the method for statistical analysis as the reviewer suggested.

In the results section, the authors should explain only their data. They should not explain how they did it (that is the method) or give their opinion (that is under discussion). Authors must rewrite this section

  • We followed the guideline of this journal below.

Results: Provide a concise and precise description of the experimental results, their interpretation as well as the experimental conclusions that can be drawn.

Authors must include limitations and conclusion sections

  • We added limitations and conclusion in the discussion section as the reviewer suggested.

The reference style is correct

Reviewer 2 Report

The manuscript is correctly structured, well written, and focused on a topic of wide interest among clinicians. Nevertheless, some minor issues have to be overcome. In introduction I would remove the sentence “Here, we demonstrated that the absence of collagen XII increased the risk of ACL 66 injury”, avoiding anticipating the take home message of the conclusion. In results a clear number of mice involved in the study is missing. In particular I would suggest specifying the size of the sample in relation to the different groups analyzed (study group Vs control group and young Vs mature mice). In this sense the numbers expressed in table 1 refers to the ACLs or to the mice? The table itself may confound the reader since it apparently regards only Col12a1 -/- reading its caption, but in the table are inserted data even about Col12a1 +/+ mice. I would eliminate figure 4 since duplicate the concept express in the text and because it is about data not statistically significant. On this field I would consider crucial the sample size in terms of strength of the analysis performed, thus it is important to provide detailed and clear numbers about ACLs and mice included in the study. Starting from your data is acceptable the statement in the discussion about considering collagen XII as a risk factor for ACL injury, but I would add in this section or in conclusion that more data are needed on that in order to be able to quantify the risk through a logistic regression analysis. This is particular important even because in mice ACL injury occurred in conditions simulating normal life activities, while in humans it occurs more frequently among athletes. In this sense the authors are invited to better justify the hypothetic mechanisms distinguishing mice from humans in terms of fragility of ACL, different biomechanics, or other, in order to support the concept that the studied animal model is reliable for risk assessment in humans.

Author Response

The manuscript is correctly structured, well written, and focused on a topic of wide interest among clinicians. Nevertheless, some minor issues have to be overcome. In introduction I would remove the sentence “Here, we demonstrated that the absence of collagen XII increased the risk of ACL 66 injury”, avoiding anticipating the take home message of the conclusion. 

  • Thank you very much for your comments. We have revised as the reviewer suggested.

In results a clear number of mice involved in the study is missing. In particular I would suggest specifying the size of the sample in relation to the different groups analyzed (study group Vs control group and young Vs mature mice). In this sense the numbers expressed in table 1 refers to the ACLs or to the mice? The table itself may confound the reader since it apparently regards only Col12a1 -/- reading its caption, but in the table are inserted data even about Col12a1 +/+ mice.

  • Thank you very much for your comments and suggestions. We included the detailed number of the mice we used in each experiment in the method section. Regarding to the table1, four wildtype and five Col12a1-/-mice were used for young stage analysis, and eight wildtype and 15 Col12a1-/- mice were used for the mature analysis. Right or left hindlimbs were blindly selected for quantitative analysis.

I would eliminate figure 4 since duplicate the concept express in the text and because it is about data not statistically significant.

  • Thank you very much for your suggestions. We understand your opinion however, we would like to keep this figure in the manuscript. Considering the increased ACL injury in mature Col12a1-/- mice, our data suggest that the age-related change including collagen XII reduction may increase risk for the ACL injury.

On this field I would consider crucial the sample size in terms of strength of the analysis performed, thus it is important to provide detailed and clear numbers about ACLs and mice included in the study. Starting from your data is acceptable the statement in the discussion about considering collagen XII as a risk factor for ACL injury, but I would add in this section or in conclusion that more data are needed on that in order to be able to quantify the risk through a logistic regression analysis. This is particular important even because in mice ACL injury occurred in conditions simulating normal life activities, while in humans it occurs more frequently among athletes. In this sense the authors are invited to better justify the hypothetic mechanisms distinguishing mice from humans in terms of fragility of ACL, different biomechanics, or other, in order to support the concept that the studied animal model is reliable for risk assessment in humans.

  • As you suggested, we increased the number of ACL discontinuity analysis particularly in mature stage. Also, we could not perform logistic regression analyses because no WT mice had ACL discontinuity and could not calculate the odds ratio. Therefore, we used Fisher’s exact test for comparison.   

Reviewer 3 Report

This article is devoted to assessing the effect of deficient Col12A1 in Collagen XII on the probability of ACL rupture. This is an actual topic, the study of which in recent years, as shown in the references, has received serious attention.

It was previously established that there is a close relationship between the Col12A1 gene polymorphism of Collagen XII and ACL rupture, but it is not known whether Collagen XII deficiency affects ACL rupture. At the same time, it is known that collagen XII is strongly expressed in tendons and ligaments and determines both the structure of the tissue and its mechanical properties, which can be significantly reduced with a deficiency of the Col12A1 gene. Therefore, Collagen XII deficiency may be a potential risk factor for the development of ACL damage and rupture [1]. Thus, prevention of ACL rupture requires an understanding of pathological mechanisms and associated risk factors leading to ACL damage.

The studies carried out by authors, results of which are presented in the article, show that Collagen XII deficiency increases the risk of ACL damage. The research was carried out at a high level. Chapter 2 briefly but convincingly describes the research methods. Results presented in Chapter 3 clearly indicate that Collagen XII deficiency causes ACL discontinuity. However, in my opinion, the authors not quite correct point out that Collagen XII is a critical component that maintains and quantitatively regulates the function of tendons and ligaments in joints. It would be more accurate to say that this is one of critical components.

Obviously, there are many variations of genetic factors that can potentially influence ACL rupture. Moreover, the relationship of these factors and their combined effect on the likelihood of ACL rupture can be very important. For example, in [2], cited by authors, it was concluded that there is a significant interaction of the Col5A1 and Col12A1 variations and their impact to risk of ACL rupture. That is, this result emphasizes the importance of studying the interaction of genes in the etiology of ACL ruptures.

This article can be published without changes. However, in order to improve it quality, it is desirable to point out the relationship between the detected risk factor for ACL rupture with other factors, and also point out the possible synergistic effect of the impact of many risk factors on the probability of ACL rupture.

  1. Baker, L.A. et al. Biologically Enhanced Genome-Wide Association Study Provides Further Evidence for Candidate Loci and Discovers Novel Loci That Influence Risk of Anterior Cruciate Ligament Rupture in a Dog Model. Frontiers in Genetics. – 05 March 2021, V.12, Art. 593515. https://doi.org/10.3389/fgene.2021.593515
  2. O’Connell, K. et al. Interactions between collagen gene variants and risk of anterior cruciate ligament rupture. European Journal of Sport Science. – 2015, Vol. 15, Issue 4. – P. 341-350. http://dx.doi.org/10.1080/17461391.2014.936324

Author Response

This article is devoted to assessing the effect of deficient Col12A1 in Collagen XII on the probability of ACL rupture. This is an actual topic, the study of which in recent years, as shown in the references, has received serious attention.

It was previously established that there is a close relationship between the Col12A1 gene polymorphism of Collagen XII and ACL rupture, but it is not known whether Collagen XII deficiency affects ACL rupture. At the same time, it is known that collagen XII is strongly expressed in tendons and ligaments and determines both the structure of the tissue and its mechanical properties, which can be significantly reduced with a deficiency of the Col12A1 gene. Therefore, Collagen XII deficiency may be a potential risk factor for the development of ACL damage and rupture [1]. Thus, prevention of ACL rupture requires an understanding of pathological mechanisms and associated risk factors leading to ACL damage.

The studies carried out by authors, results of which are presented in the article, show that Collagen XII deficiency increases the risk of ACL damage. The research was carried out at a high level. Chapter 2 briefly but convincingly describes the research methods. Results presented in Chapter 3 clearly indicate that Collagen XII deficiency causes ACL discontinuity. However, in my opinion, the authors not quite correct point out that Collagen XII is a critical component that maintains and quantitatively regulates the function of tendons and ligaments in joints. It would be more accurate to say that this is one of critical components.

  • Thank you very much for your comments and suggestions. To clarify our finding, we added our interpretation in chapter 3.

Obviously, there are many variations of genetic factors that can potentially influence ACL rupture. Moreover, the relationship of these factors and their combined effect on the likelihood of ACL rupture can be very important. For example, in [2], cited by authors, it was concluded that there is a significant interaction of the Col5A1 and Col12A1 variations and their impact to risk of ACL rupture. That is, this result emphasizes the importance of studying the interaction of genes in the etiology of ACL ruptures.

This article can be published without changes. However, in order to improve it quality, it is desirable to point out the relationship between the detected risk factor for ACL rupture with other factors, and also point out the possible synergistic effect of the impact of many risk factors on the probability of ACL rupture.

  • Thank you very much for your comments. As the reviewer suggested, we added our interpretation regarding to the relationship between collagen XII and the other potential risk factors for ACL injury that have been found in the previous studies.

Round 2

Reviewer 1 Report

The authors have made a great effort to improve the manuscript
They have answered all the comments